

# Cross-coupling polymerization at iodophenyl thin films prepared by spontaneous grafting of a diazonium salt

Nicholas Marshall[1] and Andres Rodriguez[2]

[1] Department of Chemistry and Physics, University of South Carolina Aiken, Aiken, SC, United States of America
[2] College of Pharmacy, Medical University of South Carolina, Charleston, SC, United States of America

## ABSTRACT

Cross-coupling at aryl halide thin films has been well-established as a technique for the surface-initiated Kumada catalyst transfer polymerization (SI-KCTP), used to produce covalently bound conjugated polymer thin films. In this work, we report that the spontaneous grafting of 4-iodobenzenediazonium tetrafluoroborate on gold substrates creates a durable iodoarene layer which is effective as a substrate for cross-coupling reactions including SI-KCTP. Using cyclic voltammetry of a surface-coupled ferrocene terminating agent, we have measured initiator surface coverage produced by oxidative addition of $Pd(t-Bu_3P)_2$ and estimated the rate constant of the termination reaction in the SI-KCTP system with 2-chloromagnesio-5-bromothiophene and $Pd(t-Bu_3P)_2$. We used this system to prepare uniform polythiophene thin films averaging 90 nm in thickness.

# INTRODUCTION

Since the development of the first polyacetylenes by Shirakawa and MacDiarmid in the 1970s (*Chiang et al., 1977*; *Chiang et al., 1978*), applications of pi-conjugated polymers (CPs) have proliferated along with a number of refined synthetic approaches for forming these polymers. CPs based on arene repeat units are the single largest category of CP with current practical applications, due to the inherent stability of the aromatic system *vs.* oxidation. To prepare polyarenes, oxidative polymerization approaches are common (*Kaloni et al., 2017*; *Niemi et al., 1992*). Oxidative polymerization is difficult to control and applicable only to electron-rich arene monomers. More recently, cross-coupling strategies have been developed (*Yamamoto, 2002*; *Heeger, 2001*), most commonly coupling of a aryl dihalide with an arene bearing two nucleophilic groups, e.g., an "AA/BB" strategy. Stille and Suzuki coupling are commonly employed in these syntheses, although others are reported (*Bao, Rogers & Katz, 1999*; *Argun et al., 2004*; *Yiu et al., 2012*). The single greatest refinement in the cross-coupling synthesis of polyarene CPs came in the mid-2000s from the Yokozawa and McCullough groups (*Yokoyama & Yokozawa, 2007*; *Sheina et al., 2004*), in which "AB" monomers were prepared by monometallation of an aryl dihalide. The most

Corresponding author
Nicholas Marshall,
nicholasm@usca.edu

**Figure 1** The Kumada catalyst-transfer polymerization (CTP) of a magnesiated dihaloarene proceeds in a chain-growth fashion, yielding a polyarene chain bound to the initiator.

common metallation approach uses magnesium-halogen exchange of an alkylmagnesium halide to prepare an arylmagnesium halide monomer. This modification yielded a dramatic improvement in polydispersity of the prepared polymers, especially polythiophenes and poly(p-phenylenes), relative to an AA/BB approach. The improvement is attributed to a change in mechanism, from a typical cross-coupling catalytic cycle to the so-called "catalyst-transfer" (CT) mechanism (*Miyakoshi, Yokoyama & Yokozawa, 2005*). In this mechanism, the zerovalent, coordinatively unsaturated transition metal does not dissociate from the growing polymer chain, but remains complexed to the pi-system (Fig. 1). Due to this complexation, the zerovalent center reacts next with the halide endgroup of *the same* polymer chain, and chain-growth, pseudoliving polymerization ensues (*Bryan & McNeil, 2013*).

The existence of this CT effect has been borne out by a number of theoretical studies (*He, Patrick & Kennepohl, 2018*), as well as exploits in which CTP is used to prepare block copolymers and end-functionalized polymers (*Zhang, Ohta & Yokozawa, 2018*; *Yokozawa & Yokoyama, 2009*; *Aplan & Gomez, 2017*). Many of the most common applications of CPs involve their deposition in the form of thin films, including uses in photovoltaics, electrochromics, and sensors. (*Marshall, Sontag & Locklin, 2011a*).

In the late 2000s, we and others used CTP to prepare conjugated polymer films grafted from a surface (SI-CTP) (*Senkovskyy et al., 2007*; *Sontag, Marshall & Locklin, 2009*; *Marshall, Sontag & Locklin, 2011b*; *Kiriy, Senkovskyy & Sommer, 2011*). Typically, this feat is accomplished by formation of a self-assembled monolayer (SAM) bearing an aryl halide, followed by reaction of the halide film surface with a zerovalent metal precatalyst to form a surface-bound organometallic complex. Silane (on oxide) or thiol (on metals) SAMs are the most common aryl halide films used. Many interesting surface structures have been prepared using this effective approach. However, limitations of SAMs exist; purification of silane materials is often problematic and the quality of the thin film varies based on

difficult-to-control factors such as moisture content (*Lessel et al., 2012*). The formation of thiol SAMs on noble metal surfaces is convenient and robust towards atmosphere and water (it is common practice for thiol SAMs to be prepared in the lab atmosphere using an ethanol/water mixture as solvent) but thiol SAMs are not especially durable. With all SAM techniques, the most effective coupling agents for SAM formation contain a long central alkyl chain which limits electronic coupling between the surface and the endgroup (*Trammell et al., 2007*), an undesirable property for electronic applications. So, we sought to develop a more convenient initiator system for a SI-CTP reaction, specifically the surface-initiated Kumada polymerization (SI-KCTP).

Reductive electrografting of aryl diazonium salts is well-established as a surface modification protocol (*Mahouche-Chergui et al., 2011*; *Assresahegn, Brousse & Belanger, 2015*). While careful controls such as use of antioxidant additives (*Anariba, DuVall & McCreery, 2003*; *Menanteau, Levillain & Breton, 2013*) or incorporation of a bulky protecting group (*Combellas et al., 2008*) are necessary to ensure formation of a well-defined monolayer, this method is known even in the absence of these precautions to form highly functionalized thin films composed of a conductive arene multilayer (*Lee, Brooksby & Downard, 2012*). While deposition of thin films from aryldiazonium salts using electrochemical reduction of the diazonium salt is a well-known technique for surface functionalization, spontaneous reaction of aryldiazonium salts with a surface is less common. However, a number of groups have explored this approach in recent years (*Stewart et al., 2004*; *Podvorica et al., 2009*; *Mesnage et al., 2012*). Terminal functional groups (as the *para* substituent) which have been deposited using a spontaneous method include $NO_2$ (*Laurentius et al., 2011*; *Cullen et al., 2012*), COOH (*Polsky et al., 2008*), *n*-alkyl (*Combellas et al., 2005a*), perfluoroalkyl (*Combellas et al., 2005b*), diazonium (*Marshall, Rodriguez & Crittenden, 2018*), and amine (*Kesavan & Abraham John, 2014*). The Tour group has prepared a great variety of conjugated linkers deposited spontaneously from organic solvents, particularly on silicon surfaces (*Kosynkin & Tour, 2001*). In particular, halide-functionalized thin films formed by spontaneous diazonium grafting from organic solutions have not been reported, and only a few (2,4,6-trichlorophenyl and 4-bromophenyl) (*Mesnage et al., 2012*) from aqueous solution.

Most work in the area of spontaneous surface modification with diazonium salts uses aqueous solutions of the salt, exploiting the inherent instability of arenediazonium salts in water due to diazonium hydrolysis to diazohydroxide compounds (*Lewis & Johnson, 1960*). Diazonium salt hydrolysis reliably forms thin films on metallic, oxide, and carbon surfaces (*Combellas et al., 2005a*; *Podvorica et al., 2009*; *Lehr et al., 2010*; *Berisha et al., 2016*). These films contain a substantial fraction of azo R-N=N-R′ linkages, and XPS evidence indicates that the aryl film is sometimes linked to metal surfaces through a nitrogen-metal bond (*Combellas et al., 2005a*). However, evidence is beginning to emerge that spontaneous deposition of diazonium salts in acetonitrile (and likely other polar aprotic solvents) on gold proceeds by a different mechanism, possibly by direct Au catalysis of C-$N_2^+$ bond cleavage yielding C-Au bonds (*Mesnage et al., 2012*). Bolstering this hypothesis, we have found in this work and past studies that arenediazonium-based films formed

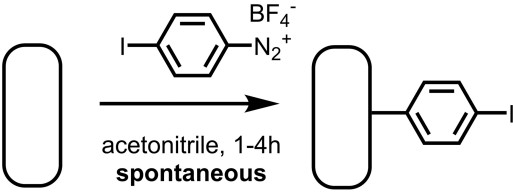

**Figure 2** The aryl diazonium salt 4-iodobenzenediazonium tetrafluoroborate reacts spontaneously with freshly cleaned Au surfaces to yield an aryl iodide-functionalized surface.

spontaneously from acetonitrile generally do not contain any nitrogen at all, in sharp contrast to aqueous-based films driven by diazohydroxide deposition.

As a component of this work, we sought to demonstrate the spontaneous grafting of the aryldiazonium halide 4-iodobenzenediazonium tetrafluoroborate, and determine whether the resulting layer reacts to form a surface-bound initiator for SI-KCTP. We found that 4-iodobenzenediazonium salt spontaneously forms a thin film at a clean gold surface, and that the resulting aryl iodide layer is convenient and effective for cross-coupling (Fig. 2). In particular, this iodoarene layer yields a high density of surface-bound Pd(II) sites active in the cross-coupling reaction as measured electrochemically using a ferrocenyl probe, $FcCH_2ThMgCl$, previously reported by our group (*Sontag et al., 2011*), and used by others for similar systems (*Youm et al., 2016*; *VonWald et al., 2018*). This surface-bound cross-coupling initiator also reacts effectively with Grignard-based thiophene AB monomers to yield polythiophene brushes.

In this work, we report a useful instance of spontaneous aryl diazonium salt grafting to a gold surface to prepare a functionalized surface which can serve as an initiator platform for the Pd-catalyzed SI-KCTP reaction (Fig. 3). XPS survey scans of the functionalized surface revealed no nitrogen in the film, supporting the hypothesis that gold can directly catalyze the dissociation of the diazonium salt to give dinitrogen. The resulting surface has a high density of reactive groups as measured under standard conditions for evaluation of SI-KCTP initiator surfaces, and a remarkably thick, brushlike polythiophene film is formed on the surface when used in the SI-KCTP reaction. Tracking of reactive group concentration during the progress of the polymerization reaction reveals that the majority of surface-bound polymer is formed after most of the endgroups have been eliminated from the surface, explaining the rough morphology of the film and possibly shedding light on similar morphology observed by other groups. The protocol we developed for these experiments uses an air-stable diazonium salt to form the aryl halide base layer and easily handled, commercially available Pd complexes to generate the surface-bound catalyst. We expect these simplifying advances in technique for this state-of-the-art conjugated polymer synthesis method to render SI-KCTP more accessible for practical applications.

**Figure 3** **Reaction of an aryl iodide layer on gold with Pd(0) followed by Grignard reagents yields a ferrocene-functionalized layer using a Fc-bearing Grignard reagent.** Use of a magnesiated dihaloarene results in a thick polythiophene film through the SI-KCTP reaction.

# MATERIALS AND METHODS

## General experimental remarks

Gold-coated slides were standard float glass microscope slides coated with roughly 100 nm Au on a Cr adhesion layer, purchased from Sigma-Aldrich. HPLC-grade acetonitrile was purchased from Alfa Aesar and stored over 4Å molecular sieves. Other chemicals were purchased directly from well-known chemical suppliers and used as received. The ferrocene coupling agent $FcCH_2ThBr$ was prepared according to the procedure freely available in the Supporting Information of our previous report (*Sontag et al., 2011*). Surface profilometry was performed at the College of Engineering and Computing, University of South Carolina Columbia, on a Veeco Dektak 3ST surface profiler using 1 mg stylus force.

1,3-bis(2,6-Diisopropylphenyl)imidazol-2-ylidene](3-chloropyridyl)palladium(II) dichloride (i-PrPEPPSI, referred to in this manuscript as PEPPSI) (*O'Brien et al., 2006*), 2,5-dibromo-3-hexylthiophene (*Buerle et al., 1993*), and 2,5-diiodothiophene (*Ebai & Marshall, 2014*) were prepared according to literature procedures. 2,5-diiodothiophene was distilled under a 5 Torr vacuum before use.

## Synthetic procedures
### *Synthesis of 4-halobenzenediazonium tetrafluoroborate salts*
4-iodoaniline was prepared by iodination of aniline with molecular iodine in aqueous sodium bicarbonate solution according to Vogel's procedure (*Vogel et al., 1996*), and recrystallized from boiling heptane to yield yellow needles. 4-bromoanilinium hydrochloride was obtained from Eastman and used as received. Recrystallized 4-iodoaniline or as-received 4-bromoanilinium hydrochloride (2.5 mmol) was suspended in 2.0 mL deionized water and cooled below 0 °C in an ice/salt bath. 2.0 mL of concentrated HCl was added, and the reaction was stirred until it had returned to the ice bath temperature. Sodium nitrite (0.19 g, 2.8 mmol) was added in a single portion, and the reaction was stirred for 30 min. After this time, sodium tetrafluoroborate (0.55 g, 5.0 mmol) was dissolved in a minimum amount of deionized water and added to the reaction while stirring. A yellow precipitate was formed and collected by vacuum filtration, and washed with cold water, cold methanol, and diethyl ether. This product was taken up in a minimum amount of acetonitrile and re-precipitated by adding diethyl ether to yield crystalline product.
4-iodobenzenediazonium tetrafluoroborate: yellow crystals, 0.093 g, 12% [1]H NMR (DMSO-d$_6$, 60 MHz): $\delta$ (ppm) 8.47 (d, $J$ = 9.4 Hz) 8.65 (d, $J$ = 9.4 Hz). [13]C NMR (acetone-d$_6$, 15 MHz): $\delta$ (ppm) 133, 141. IR (ATR, cm-1): 3373 (w), 3099 (m), 2997 (w), 2282 (s, NN triple bond stretch), 1548 (m), 1461, (m) 1406, (m) 1314, (m) 1289, (m), 1123 (w), 1021 (br), 824, 768 (m).

4-bromobenzenediazonium tetrafluoroborate: white crystals, [1]H NMR (acetone-d$_6$, 60 MHz): $\delta$ (ppm) 8.24 (d, $J$ = 8.9 Hz) 8.69 (d, $J$ = 9.0 Hz). [13]C NMR (acetone-d$_6$, 15 MHz): $\delta$ (ppm) 134, 135. 0.303 g, 45%. Aromatic carbons without H atoms are not typically observed in 13C NMR using the Anasazi 60 MHz NMR.

### Deposition of aryl iodide thin films

A 50 mM solution of freshly recrystallized 4-iodobenzenediazonium tetrafluoroborate was prepared in dry acetonitrile, typically a 10 mL sample in a clean 20 mL scintillation vial. 1–3 pieces of cut gold-coated glass slide were cleaned. Cleaning included sonication in isopropyl alcohol, washing with deionized water, followed by 5 min immersion in freshly prepared sulfuric acid/30% hydrogen peroxide "piranha" solution, 3:1 v:v. (Caution: this mixture is violently reactive, especially with organic materials; the smallest amount practical should be prepared, and piranha should be handled using exclusively glass materials in a clean, uncluttered hood. The operator should wear body protection, gloves, and face protection.) Freshly cleaned slide substrates were blown free of moisture in a stream of nitrogen gas, and added immediately to a freshly prepared iodobenzenediazonium salt solution. Substrates were allowed to stand without stirring for 1–2 h, then removed and washed with copious amounts of acetonitrile and dried in a stream of nitrogen gas. Functionalized gold substrates were stored in a covered Petri dish in lab atmosphere with no additional precautions until use.

### Surface coupling procedure

In a nitrogen-filled glove box, 10 mM solutions of Pd initiators were prepared in toluene. Pd(t-Bu$_3$P)$_2$ solution was prepared directly from the solid, available commercially from TCI as a beige powder and yielding a faintly yellow solution. The air-stable Pd(II) complex i-Pr-PEPPSI was suspended in toluene in an amount calculated to yield a 10 mM solution, and 2 equiv. freshly titrated i-PrMgCl in 2-MeTHF was added dropwise while agitating. A clear yellow solution was obtained, which gave a grey precipitate after a few minutes. This solution was used as formed.

Grignard coupling agents were prepared by addition at room temperature while stirring in the glove box of 1.0 equivalent of freshly titrated i-PrMgCl.LiCl solution to 10 mL of a 50 mmol solution of the aryl halide (2,5-dibromothiophene or FcCH$_2$ThBr) in THF. Aliquots of the polymerization solution were saved, removed from the glove box, and quenched with water to confirm the presence of excess halide; excess i-PrMgCl is detrimental to the cross-coupling reaction.

Aryl iodide functionalized slides were placed in a Pd initiator solution within the glove box and the solution heated to 60 °C on a hot plate in a sealed vial for 2 h. Slides were removed from the vial, washed 3× with toluene, and transferred to a 50 mM solution of Grignard reactant prepared as above (either 2-chloromagnesio-5-bromothiophene or
ferrocene probe ClMgTh-CH$_2$-Fc) and heated to 60 °C overnight. After reaction, samples were removed from the glove box, exposed to atmosphere, and washed with THF, ethanol, water, and acetone. For samples termed "rigorously cleaned," PT-grafted films were briefly sonicated (ca. 30s) in acetone and were transferred to a scintillation vial half filled with chloroform, heated to boiling for several minutes, and allowed to cool. This cycle was repeated twice. The samples were removed from the film and dried in air.

## Electrochemistry

Cyclic voltammetry (CV) and alternating-current voltammetry were performed using a BASi PalmSens 3 potentiostat/galvanostat in a three-electrode cell configuration. A 100 mM solution of tetrabutylammonium hexafluorophosphate in dichloromethane was used as electrolyte. A silver wire pseudoreference electrode was used for which the Fc/Fc$^+$ redox couple on gold appeared at 0.489V, and a platinum wire was used as the counter electrode. Electrochemical surface coverages were determined by direct integration of CV peaks, and are reported as an average of anodic and cathodic peaks. Surface area of electrodes was estimated by imaging of the electrode surface with a 2.83 cm$^2$ standard and image analysis using ImageJ to determine working area (*Schindelin et al., 2012*).

## X-ray photoelectron spectroscopy
### General instrumental information

XPS measurements were performed in the USC Center for Engineering and Computing. The instrument used was a Kratos Axis Ultra DLD with a hemispherical analyzer and monochromator-equipped Al K$\alpha$ source. The Au 4f$_{7/2}$ peak was used as a binding energy reference at 84.0 eV, with spectra corrected so that this reference peak appeared at the standard value.

### XPS thickness estimates

For film thickness estimates using XPS, we assume that Au is covered by an homogeneous carbon layer in order to calculate the Inelastic Mean Free Path (IMFP) of photoelectrons through the deposited layer. For photoelectrons with kinetic energy higher than 150 eV a good approximation for the IMFP ($\lambda$) in nanometers is $\lambda = BE^{1/2}$, where B has a numerical value of 0.087 for organic layers and E is the photoelectron kinetic energy. Using a Al K$\alpha$ excitation photon beam, kinetic energies for the C 1s and Au 4f electrons are 1202 and 1402 eV respectively. Experimental XPS peak intensities, including the attenuation of Au signal by C layer, are described by the following equations:

(1) $I_C = I_C(\infty) \times (1 - e^{d_C/\lambda_C \, cos(\theta)})$

(2) $I_{Au} = I_{Au}(0) \times (1 - e^{d_C/\lambda_{Au} \, cos(\theta)})$

where I$_C$ and I$_{Au}$ are the experimental intensities, I$_C(\infty)$ = 0.25 and I$_{Au}(0)$ = 4.95 are atomic sensitivity factors, $\theta = 0$ is the angle between the electron analyzer entrance and the surface normal of the analysis sample, and d$_C$ is the thickness of the carbon overlayer being modeled.

### Scanning electron microscopy

SEM and EDX were performed on an Hitachi Cold Field Emission 8200 Series FE-SEM at the Applied Research Center, Aiken, SC, USA. A 9 kV accelerating voltage was used for typical images shown.

### Vibrational and NMR spectroscopy

A Thermo Electron Nicolet 4700 Fourier transform IR spectrometer using a DTGS detector and fitted with a grazing angle accessory was used to collect IR spectra of thin films. Bulk IR spectra of samples were collected on a Nicolet 380 spectrometer with a Smart Orbit diamond ATR attachment. 64 scans were summed at a four $cm^{-1}$ resolution to produce a typical spectrum.

Routine 1H and 13C NMR of monomers and initiators were taken using a Anasazi EFT 60 MHz FT-NMR spectrometer referenced to tetramethylsilane at 0 ppm.

## RESULTS

### Preparation of aryl iodide surface: characterization of the aryl iodide pre-initiator layer

X-ray photoelectron spectroscopy of the aryl iodide-treated surface gives a satisfactory picture of its structure. Analysis of the intensity ratios of the Au 4f and C1s peaks (Fig. 4) yields an estimated thickness of the thin film of 2.5 nm, consistent with other reported spontaneously grafted thin films (*Cullen et al., 2012*; *Kesavan & Abraham John, 2014*) and roughly half the thickness of reductively electrografted iodophenyl films (*Touzé et al., 2019*; *Müri et al., 2011*). The signature of iodine is naturally easy to observe in XPS. After correction, the film gives a 1:8 I:C atomic ratio, when in an ideal monolayer 1:5 is expected. The shape of the C1s C-C peak is consistent with the presence of 2 $C_{sp^2}$ species, with the higher-binding-energy, lower-intensity peak representing the C-I species. The spectrum is consistent with that reported for cathodically electrografted 4-iodobenzenediazonium salt (*López, Dabos-Seignon & Breton, 2019*).

### Initiator density estimation using cross-coupling of a ferrocene probe

While a halide-functionalized polyarene layer may have many applications (*Touzé et al., 2019*; *Müri et al., 2011*), we developed this process as an easy entry to surface-directed cross-coupling reactions. We can estimate an upper bound on the efficiency of the oxidative addition step (Fig. 3) by reaction of a Pd(0) complex followed by cross-coupling with a ferrocene probe. We have previously shown that Pd(t-Bu$_3$P)$_2$, developed by Fu and co-workers (*Littke, 2002*), is an effective Pd(0) precatalyst for forming surface-bound initiators for Kumada catalyst transfer polymerization through oxidative addition. Reaction of this precatalyst with our spontaneously grafted aryl iodide layer, followed by quenching with the ferrocene-bearing Grignard probe species FcCH$_2$ThMgCl, gave a ferrocene-coated gold surface. Using cyclic voltammetry (CV) (Fig. 5), the surface coverage of ferrocene was estimated to be $\Gamma = 2.8 \times 10^{-10}$ mol/cm$^2$.

An important ancillary finding of this work is that the standard procedure (*Sontag et al., 2011*; *Youm et al., 2016*; *VonWald et al., 2018*) for forming FcCH$_2$ThMgCl, using

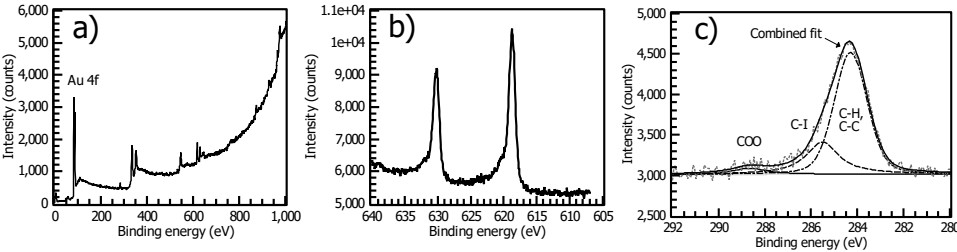

**Figure 4** **XPS of the spontaneously deposited 4-iodophenyl layer.** (A) Survey scan of the film reveals no nitrogen near 400 eV, and a strong Au 4f peak indicating a thin (ca. 2.5 nm) organic film. (B) I 3d XPS of the film gives a strong signal. (C) The breadth of the C 1s peak is consistent with two major species present.

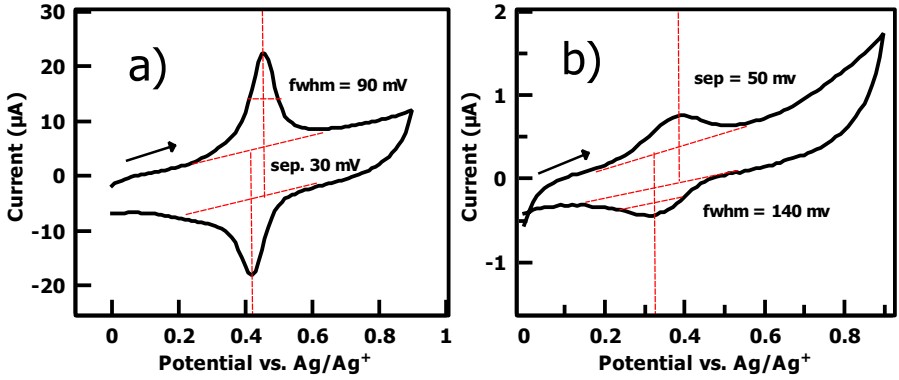

**Figure 5** (A) Cyclic voltammetry of PhI thin film on gold (0.1M TBAPF$_6$/DCM solution, 100 mV/s scan rate, Ag wire reference electrode, second cycle shown) after reaction with the ferrocene cross-coupling probe Fc-CH$_2$-ThMgCl shows a densely packed surface ($2.8 \times 10^{-10}$ mol/cm$^2$) and a near-ideal surface redox couple consistent with a thin arene layer. (B) A commercial bromophenyl silane on ITO yields an order of magnitude lower surface coverage, larger FWHM of redox peaks, and larger peak-to-peak separation under the same electrochemical conditions.

1 equivalent of isopropylmagnesium chloride (iPrMgCl), does not completely convert the precursor halide into the probe species FcCH$_2$ThMgCl (Fig. S4). While the use of 3 equivalents of iPrMgCl does accomplish this conversion, the resulting Grignard solution yields a lower surface coverage after cross-coupling than the partially converted material (Fig. S5). Groups using this procedure should be aware that this complication exists, and surface coverage values generated using this protocol should be regarded as lower bounds.

## SI-KCTP using the surface-bound initiator

Reaction of species of the form X-Ar-MgX' with surface-bound Pd(II) and Ni(II) catalysts (the SI-KCTP reaction) has been established as a state-of-the-art technique for surface-initiated chain-growth polymerization yielding polyarene films, including polythiophene films (*Marshall, Sontag & Locklin, 2011b*; *Bryan & McNeil, 2013*; *Neo et al., 2016*). The surface-bound Ar-PdL$_2$I complex produced from our film's reaction with the Fu catalyst

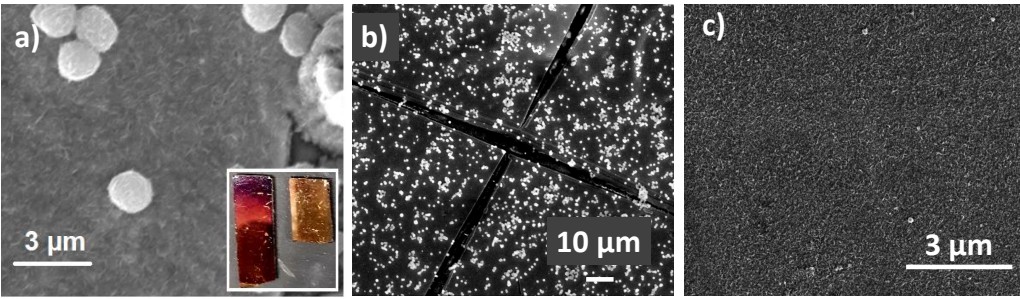

**Figure 6** (A) SEM of a PT film produced by SI-KCTP according to **Fig. 2** shows a morphology consistenting of a rough PT coating. Physisorbed PT particles are also visible. Inset: The PT film produced on gold is a dark purple color, indicating closely packed PT chains. (B) A lower-magnification image of the same PT film (in which a crack in the metal coating is also visible) reveals a regular scattering of narrow-size-distribution (**Fig. S7**) PT particles. EDS mapping of this region (**Fig. S6**) shows carbon and sulfur signals associated with the film and particles but not the glass below, confirming the presence of polythiophenes specifically coating the gold surface. (C) Rigorous cleaning removes physisorbed polythiophene, but leaves the surface-bound PT brush (roughly 90 nm by profilometry) intact (**Fig. S2**).

is effective in the SI-KCTP reaction, yielding a thick (avg. 90 nm) PT film with a purple color, visible as a rough coating on gold in scanning electron microscopy (SEM) (Fig. 6). In the polymerization reaction with the Fu catalyst, no precipitate of polymer was observed in solution after several hours. On the other hand, the surface-bound initiator produced from the reduced NHC-Pd catalyst PEPPSI (1,3-bis(2,6-diisopropylphenyl)imidazol-2-ylidene](3-chloropyridyl)palladium(II) dichloride) reacted within 15 min to yield visible red polymer suspended in the reaction vial. This sample was coated with a thinner, red polythiophene film with the typical bumpy morphology of an unsubstituted conjugated polymer (*Sontag, Marshall & Locklin, 2009*) after reaction was complete (Fig. S3). Reflectance UV-visible spectroscopy of the films is consistent with the presence of a thinner PT film produced by PEPPSI (Fig. 7). The regular decoration of physisorbed PT nanoparticles visible in the post-polymerization film (Fig. 6A) was an unexpected discovery, since the polymer film was washed well with water and organic solvents after the reaction was complete.

XPS of the Fu and PEPPSI PT films (Fig. 8) after reaction reveal several critical aspects of the SI-KCTP reaction. First, both samples give spectra consistent with the presence of thick polythiophene films, with the Au peak largely effaced due to the thick covering layer. The relative ratios of C to S are not significant here due to the disproportionate effect of surface adventitious carbon as in the initiator layer (Fig. 4) but the much larger sulfur signal in the Fu film is certainly due to this film's greater thickness, consistent with UV-Vis and IR measurements (Fig. 7). Since we can tell from the S signal as well as IR and absorption spectroscopy that the Fu film is thicker, Au signal (roughly equal in both samples) is likely due to defects in the film, rendering the thickness fitting procedure used for the initiator layer inapplicable here (*Wojdyr, 2010*). Fits of the C 1s region of both films (Fig. S13) confirm the larger proportion of the $C_{sp^2}$ component of the Fu film, consistent with this film's greater thickness.

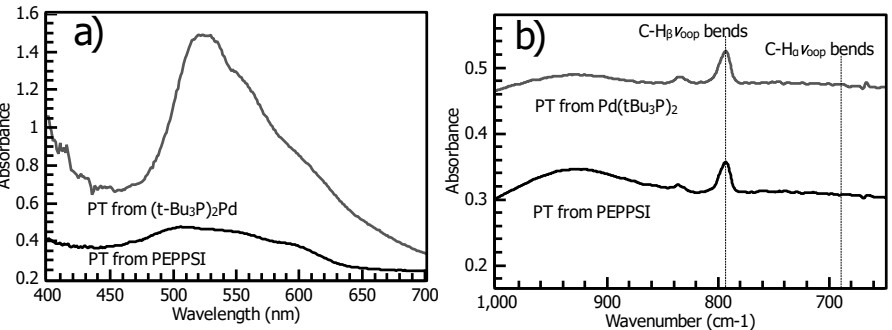

**Figure 7** **(A) Reflectance UV-vis of polythiophene films prepared using (t-Bu₃P)₂Pd ("PT-Fu") and PEPPSI Pd ("PT-PEPPSI") catalysts. The Fu catalyst yields a thick, densely grafted film. (B) Grazing-angle IR spectroscopy of PT-Fu and PT-PEPPSI films reveals the distinctive out-of-plane C-H bending mode of polythiophenes.** In both films, little C-H$_\alpha$ is observed, though it is slightly stronger in the PT-Fu film.

## Quenching studies of the growing Fu-PT film

We characterized the film growth and catalyst activity as a function of time by quenching a series of growing Fu PT films with FcCH₂ThMgCl. Ferrocene surface coverage in the films after quenching gives a reasonably direct measure of active chain ends at the time of quenching (Fig. 9). Formation of these ferrocene-terminated PT films allows characterization of the PT films by alternating-current voltammetry (ACV), revealing the gradual evolution of $k_{ET}$ in the growing film (Fig. 10). Each timepoint was also characterized by IR and reflectance UV-vis. (Fig. 11).

We attempted the SI-KCTP procedure with two other common monomers, 2,5-diiodobenzene and 2,5-dibromo-3-hexylthiophene, and repeated the preparation of the Fu film with 4-bromobenzenediazonium tetrafluoroborate used to form the initiator layer (Figs. S9 and S10). Magnesiation and polymerization of 2,5-diiodothiophene for 12 h yielded a thick (70 nm) polythiophene film which was measurably more uniform (rms roughness of 31 nm) by stylus profilometry than the film produced by the dibromothiophene monomer (42 nm) (Figs. S14 and S15). 2,5-dibromo-3-hexylthiophene did not yield a film of P3HT detectable at 12 h by reflectance UV/Vis, but did give enough grafted material to be detectable by IR (Fig. S17). On the other hand, a gold surface prepared with a C18 SAM gave no substantial film of PT by UV-vis, an important negative control (Fig. S11).

## DISCUSSION

### Initiator structure and surface coverage

Briefly, XPS of the spontaneously grafted film is consistent with a thin layer comprised primarily of C$_{sp^2}$ atoms with some adventitious carbon, including residual carbonate from oxidative cleaning (Fig. 4C). This adventitious material, which is practically inescapable in samples exposed to the air and is present in a freshly cleaned "bare" gold substrate (Fig.

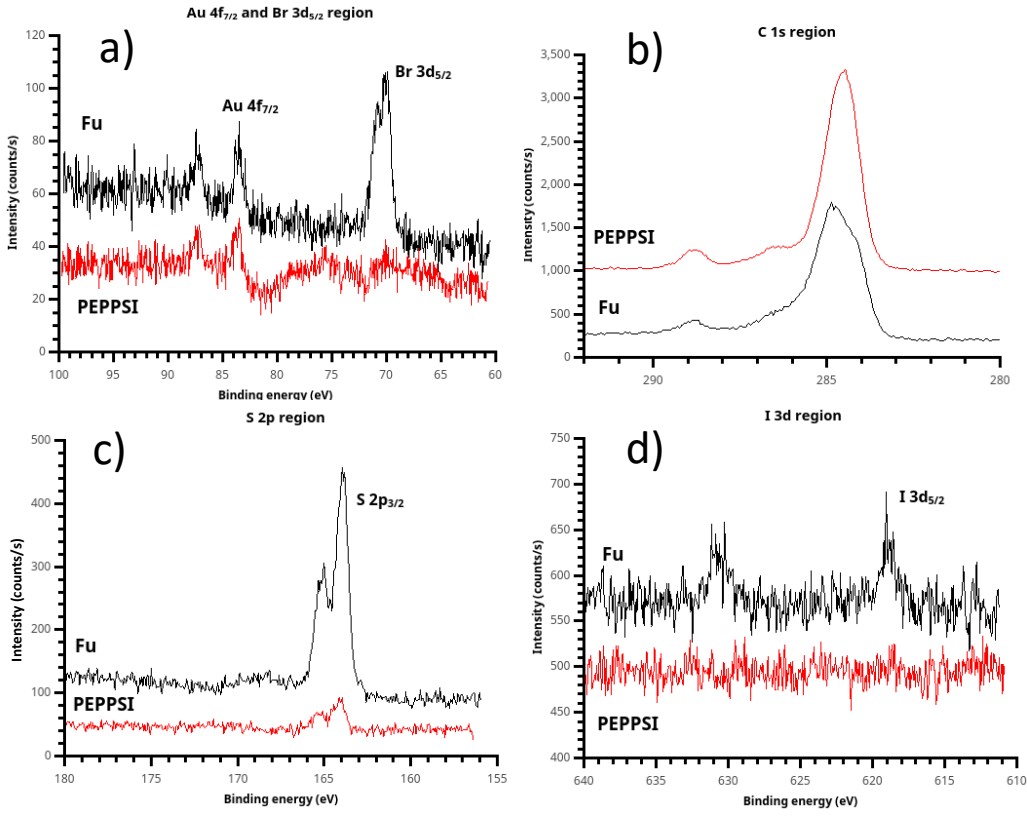

**Figure 8 X-ray photoelectron spectroscopy (XPS) of polythiophene films prepared using (t-Bu₃P)₂Pd ("Fu film") and PEPPSI Pd ("PEPPSI film") catalysts.** (A) Some bromine chain ends are present in the Fu film, implying that termination can occur by catalyst dissociation from the growing chain. Br is not observed in the PEPPSI film; therefore, termination must occur by disproportionation. (B) Due to the substantial (ca. 80 nm) thickness of the Fu film, the $C_{sp^2}$ component is clearly visible in its C 1s spectrum. Higher binding energy peaks are also visible due to atmospheric oxidation of the polymer, yielding COO and C-O groups. (C) The S 2p peak, near 165 eV and consistent with a thiophene sulfur, is very strong in the Fu film. (D) Again PEPPSI shows its reactivity, with the iodine initiator group completely eliminated from the PEPPSI film. A faint but recognizable iodine signal is seen in the Fu film, showing that oxidative addition of the Pd(0) catalyst is not 100% efficient despite high observed electrochemical surface coverage.

S12), explains the deviation from ideal I:C ratio and the presence of the C=O peak near 288.5 eV.

The observed surface coverage for the Pd initiator on a spontaneously grafted iodophenyl layer is identical to that determined by the Tour group for the ferrocene surface coverage of a directly electrografted ferrocenyl arenediazonium salt on ITO (*Lu, He & Tour, 2008*) and is not significantly different from the reported ($\Gamma = 3.0 \times 10^{-10}$ mol/cm²) achieved by carbodiimide coupling of ferrocenecarboxylic acid to a amino-terminated thiol SAM (*Seo, Jeon & Yoo, 2004*). CV of the cross-coupled ferrocene layer gives a very low (30 mV) separation between anodic and cathodic peaks consistent with a fast, ideal surface-confined redox reaction. The full width at half max (FWHM) of each peak is 90 mV, indistinguishable from the Nernstian ideal value of 90.6 mV. It is worth noting that unlike many observed close-packed monolayers, this CV does not display broadening of the peak due to repulsive

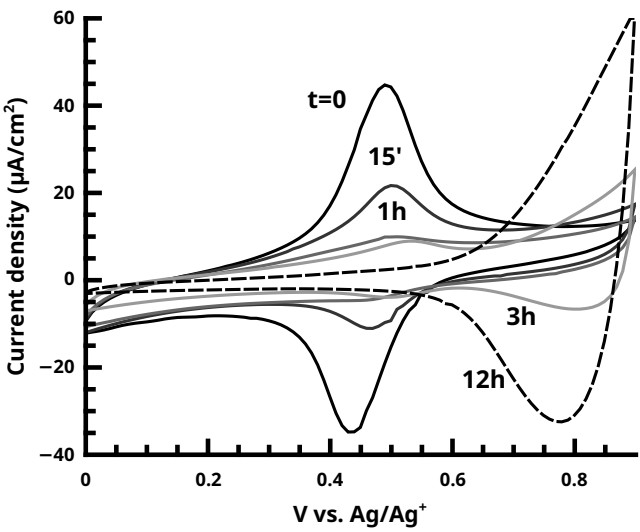

**Figure 9** **The disappearance of the ferrocene endgroup over time in quenching studies of the active SI-KCTP reaction with Fu catalyst is monitored by CV.** At longer times (3 h and 12 h curves) doping/de-doping waves of PT begin to become visible with oxidation onset ca. 0.6 V.

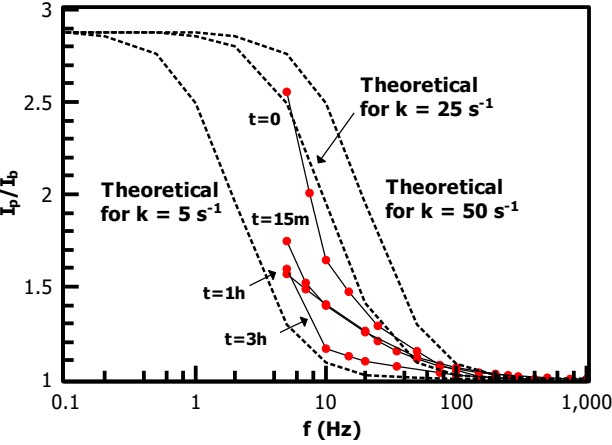

**Figure 10** **Alternating-current voltammetry of the ferrocene-terminated films allows an estimate of redox rate constant values $k_{ET}$.** Values range from ca. 25 s$^{-1}$ (FcCH$_2$ThMgCl grafted directly to IPh with no PT) to 5 s$^{-1}$ (after 3 h polymerization time).

interactions between the redox sites (*Vogel et al., 2017*). This observation is consistent with our proposed structure for the iodophenyl surface, in which the reactive iodine groups are distributed over a thin, disordered 3-dimensional polyphenylene film rather than packed more closely in a true 2-dimensional SAM.

For contrast, we performed the same coupling reaction using an indium tin oxide substrate functionalized with the commercially available silane 4-bromophenyltrimethoxysilane (Fig. 5). The surface coverage of the resulting ferrocene layer was significantly lower, (ca. $1 \times 10^{-11}$ mol/cm$^2$, Fig. 5) and the 50 mV peak-to-peak

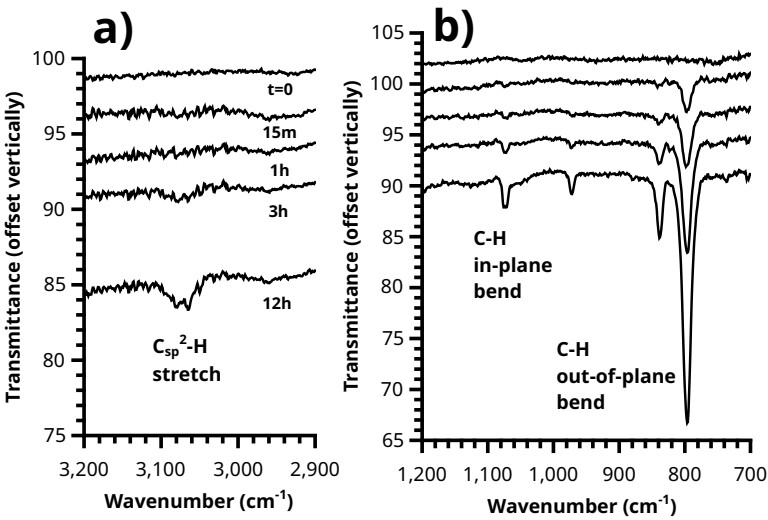

**Figure 11** Growth of the polythiophene brush over time can be plainly observed in the ATR-IR spectrum, with (A) C-H stretching and (B) out-of-plane / in-plane C-H bending peaks visible.

separation of the redox waves in the silane-based film indicates a slower rate of the redox reaction *vs.* the iodophenyl surface, which displays only a 30 mV separation. Downard has correlated layer thickness of nitrophenyl groups with $NO_2$ group concentration in spontaneously deposited nitrophenyl films, giving concentrations of $6 - 14 \times 10^{-10}$ mol/cm$^2$ (*Lehr et al., 2009*). Surprisingly, our electrochemically measured surface coverages for the iodophenyl layer based on cross-coupling with the ferrocene probe are reasonably consistent with these values, despite the much greater steric demands of the cross-coupling reaction compared to the electrochemical reduction of a -$NO_2$ group. Overall, cyclic voltammetry of the cross-coupled aryl iodide substrate indicates efficient conversion of surface aryl iodide groups in the arenediazonium-based film to Ar-Pd(II) groups and subsequent reaction with an aryl Grignard reagent in solution, effectively proving the formation of a surface-bound Pd(II) catalyst for Kumada coupling.

## Insights into PT structure and catalyst fate from absorption spectroscopy and XPS

The film produced by the Fu catalyst (PT-Fu) has an intense and narrow visible-spectrum transition redshifted relative to the fundamental absorption of the PEPPSI film, conferring a purple color on the film and indicating in this case that the chains are sufficiently close in origin point to be forced into an ordered configuration with a high conjugation length (*Mitchell, 2007*). This observation is consistent with the surface coverage observed by ferrocene probe, which matches values for a close-packed monolayer (Fig. 5). Additionally, copious amounts of PT are observed in solution with the PEPPSI catalyst, but little or none with the Fu catalyst. These observations are consistent with the polymerization reaction catalyzed by PEPPSI complex having a much higher reaction rate than the Fu catalyst, but the surface-confined reaction with PEPPSI undergoing rapid termination such that the catalyst soon dissociates, ending surface-directed chain growth of the polymer brush. The

high conjugation length in the Fu film may be primarily due to increased grafting density relative to the PEPPSI film and to literature reports of thinner unsubstituted PT films (*Chatterjee et al., 2018*). Thick polythiophene films prepared by cross-coupling methods have been reported to form ordered structures, yielding a similar sharp fundamental transition resembling the visible spectrum of monodisperse solution polymers (*Fuks-Janczarek et al., 2006*) and vibronic structures tailing off into the near-infrared (*Youm et al., 2016*). Ordered structure in other polythiophenes has been reported to yield a purple color (*Mitchell, 2007*), similar to our own Fu film (Fig. 6, inset).

A reasonable explanation for the interesting 500 nm diameter low-polydispersity (Fig. S7) features is that they were formed from free Pd catalyst centers, not bound to the surface, which were present during the reaction. Such centers could have been due to (1) physisorbed Pd(0) not removed by washing after catalyst deposition, (2) chain transfer of Pd(0) off the growing chain during the reaction, or (3) detachment of the growing chain from the surface at the original point of attachment (i.e., desorption of an initiator phenyl group after the reaction has started). The presence of these catalyst centers, through whatever means, resulted in the formation of solution polymer. As the polymer grew, the solubility and swellability of the chain decreased quickly, and aggregates formed, eventually physisorbing onto the surface. Attributing these nanoparticles to loose Pd is consistent with our observation that the nanoparticles give a strong palladium signal as observed by energy-dispersive X-ray spectroscopy (EDS) (Fig. S1), and the observation that a small amount of polythiophene is detectable by reflectance UV-vis even in the control slide coated only with an alkanethiol SAM (Fig. S11). Relatively few SI-KCTP films in the literature have been characterized by SEM such that the micron-scale morphology is known, and aggregated solution polymer features may be relatively common in low-solubility polyarene films prepared by SI-KCTP. The particles remain physisorbed after rinsing with water and acetone to quench the polymerization, but a rigorous cleaning by sonication in acetone followed by exhaustive extraction in boiling chloroform removed the particles, leaving the film intact.

Examination of the halogen signals in XPS yields insight into the course of the polymerization. The PEPPSI-treated film is completely scoured free of both iodine and bromine signals. The absence of iodine strongly suggests that the Pd(0) active species in the PEPPSI-catalyzed polymerization undergoes oxidative addition very efficiently, since oxidative addition of Pd(0) is the likeliest mechanism by which iodine can be eliminated from the initiator. The complete absence of bromine in the PEPPSI film is equally suggestive. This result implies that dissociation of Pd(0) during the catalyst-transfer step (Fig. 1) must *not* be the means of chain termination in the PEPPSI-catalyzed reaction. Dissociation would leave bromine-terminated chains. The other possibility for termination is by disproportionation of Pd(0) catalyst ends. This process would lead to C-terminated, looped brushes, consistent with Nesterov's result from Ni(0) catalyzed PT formation studied by small-angle neutron scattering (*Chatterjee et al., 2018*). Alternatively, the surface-confined chain-growth polymerization could simply be slow, such that active Pd(II) endgroups remained on the film and resulted in H-terminated chains on exposure to water during cleaning. We view this possibility as unlikely, due to (a) the fast production

of copious amounts of solution polymer we observed during the reaction and (b) the fact that, unlike the Fu PT film, we did not observe residual Pd in the PEPPSI PT film by SEM/EDS. Based on the XPS results, we believe that the PEPPSI catalyst gives a very active Pd(0) species which undergoes oxidative addition rapidly but also terminates rapidly by disproportionation, yielding mostly solution polymer.

A small amount of residual iodine is visible in the Fu film, indicating that the oxidative addition step is not 100% efficient for this catalyst. However, the iodine signal is dramatically decreased *vs.* the initiator layer, (Fig. 4) so that this result is not necessarily inconsistent with the high surface coverage observed by ferrocene probe (Fig. 5). A substantial bromine signal is seen in the Fu film, indicating that dissociation is a significant termination process in the growing Fu film, although not ruling out disproportionation and the presence of still-active chain ends as alternative fates for individual chains.

## Polymerization, redox, and termination kinetics from quenching studies

Quenching of growing PT films with ferrocenyl Grignard yields insights into kinetics of both the polymerization reaction and the termination reaction(s). The induction period seen in Fig. 12 is particularly interesting. Why does this feature appear? Comparison of these data to the evolution of the IR spectrum over time (Fig. 11) reveals a likely answer.

The induction period is not observed in IR. Rather, the intensity of in-plane and out-of-plane thienyl C-H bending modes grows smoothly starting at $t = 0$. A reasonable interpretation of these two observations is that the growing polymer chains at t = 15′ and t = 1 h have not reached the length required to fold into ordered structures (*Youm et al., 2016*) which yield the consistent conjugation length (about $n = 8$) giving rise to the absorption at 480 nm and the optical band gap (matching literature for polythiophene per (*Kaloni, Schreckenbach & Freund, 2016*) near 1.9 eV (Fig. S16). This interpretation is also consistent with voltammetric measurements of these samples, where only the 3 h and 12 h samples show the doping/dedoping process of polythiophene, (Fig. 9) and its associated electrochromism (Video S1, 12 h brush shown). A similar induction period has been observed in SI-KCTP formation of poly(3-methylthiophene) with a Ni catalyst (*Doubina et al., 2012*).

One very clear conclusion from comparison of the active endgroup concentration to the polymer thickness as each evolves over time is that the majority of catalyst centers are removed by termination *before* an appreciable thickness of polymer has formed (Fig. 12). This result implies that the observed film thicknesses are the product of an initiator density significantly lower than that measured by cross-coupling probes of the initiator surface alone. A perfectly uniform and brushlike surface would be very smooth (*Pyun, Kowalewski & Matyjaszewski, 2005*), and a surface coverage of actual polymer chains equal to the initial measured initiator surface coverage (Fig. 5) would be sufficiently closely packed to force a brushlike structure. Without knowledge of the observed reduction in initiator surface coverage, it would be difficult to explain the observed fibrillar or nodular morphology of the surface, which is found in *all* reported nanoscale imaging of surface-bound conjugated polymers lacking solubilizing sidechains. We interpret individual fibrils or nodules as

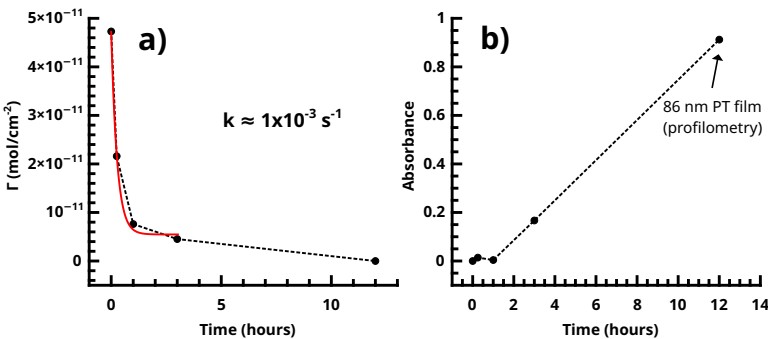

**Figure 12** (A) Dissociation of the Pd(0) species in the SI-KCTP reaction with the Fu catalyst takes place with *k* on the order of $10^{-3}$ s$^{-1}$. (B) Tracking absorbance at 480 nm is a good measure of growth of the polythiophene film, which takes place at roughly 10–11 nm/h after an induction period corresponding to a shorter conjugation length.

features formed by chains grown at the longest-surviving catalyst sites. Typical feature sizes (as measured by SEM and rms roughness of stylus profilometry, Figs. S14–S15) are 30–50 nm. An 50 nm PT feature must have a degree of polymerization (DP) of roughly 130 using a rigid-rod model, or roughly 200 using a wormlike chain model with a 12–15 nm persistence length. This reasonably high molecular weight is not inconsistent with FTIR measurements. The degree of polymerization in polythiophene can be estimated from IR (*Furukawa, Akimoto & Harada, 1987*) by fitting the ratio of absorbances at 670 and 790 cm$^{-1}$ (corresponding to endgroup and repeat unit out-of-plane C-H bending modes respectively,) but the endgroup peak is not visible in the Fu and PEPPSI films with our instrument. To be indistinguishable from noise in our system, the absorbance value must be <0.001, (Fig. 7) giving a lower bound of DP = 56 by comparison to the observed repeat unit out-of-plane bend intensity of 0.05.

The low-frequency regions of the ACV data are not well enough defined to extract a precise measurement of $k_{ET}$ for the Fc-functionalized films, but the data allows a reasonable overview of the trend in the rate constant in the growing film. The substrate consisting of FcCH$_2$ThMgCl coupled directly to the 2.5 nm iodoarene layer has $k_{ET}$ near 25 s$^{-1}$, while the 3 h sample is bounded by $k_{ET}$ = 5 s$^{-1}$. These values are comparable to long-chain alkylthiol-based ferrocene monolayers (*Eckermann et al., 2010*), despite the iodoaryl/PT film's several-fold greater thickness. ACV of the 1 h film, in particular, shows a relatively low slope at low frequencies, suggesting that the film is made up of a heterogeneous collection of redox sites with different individual $k_{ET}$ values (ibid.) (*Chidsey, 1991*). The slow decrease of observed rate constants as the film grows is reasonable in that polythiophene is a conductive polymer and becomes doped by oxygen in a matter of a few minutes (*Cook, Furube & Katoh, 2012*), providing a hopping-based conductivity mechanism to allow communication of electroactive endgroups with the electrode.

The likeliest interpretation of the results from attempted 3-HT polymerization is that termination processes were complete in the "induction period" (Fig. 12) before a sufficiently large fraction of surface chains had grown to a conjugation length sufficient to yield the

characteristic absorption of a polythiophene. This result is consistent with our previous finding that monomers with bulky groups on the monomer do not polymerize well in the SI-KCTP reaction using SAM-based initiators even in cases which are known to work in solution (*Marshall, Sontag & Locklin, 2011b*). This limitation has been overcome by the use of grafted macroinitiator layers (*Kiriy, Senkovskyy & Sommer, 2011*) or by initiating the polymerization from a highly curved, colloidal nanoparticle rather than a flat, monolithic surface. Based on our results, it seems that the steric restriction of the diazonium-based arene layer is similar enough to that of a SAM to result in the same limitation on SI-KCTP reactions. The coupling of ferrocene probe to the bromoarene layer was not as efficient as that observed in the iodoarene layer, giving roughly 10% of the coverage (and a non-ideal voltammogram) compared to iodophenyl under the same conditions (Figs. S10 and S12). However, a substantial polythiophene film was still produced with the bromophenyl initiator surface, ca. 20 nm as roughly estimated by comparison of the UV-Vis absorbance to that of the known brush. This result is reasonable, given our earlier finding that a relatively small fraction of the original catalyst endgroups are responsible for the majority of polymer production.

Overall, microscopy, voltammetry, and spectroscopy of substrates produced from the reaction of polyarene films with Pd(0) complexes followed by BrThMgCl reveals a roughly 90 nm thick, regular film of polythiophene, consistent in morphology with other films of polythiophenes with small or no side chains reported in the literature (*VonWald et al., 2018*), and which is robust to rigorous cleaning procedures. In our experience, the use of the commercially available Fu Pd catalyst is very convenient and reproducible relative to approaches based on Ni(COD)$_2$ with various ligands, and the surface coverages attained by our method are as good or better than other state-of-the-art techniques (*Youm et al., 2016*; *Sontag et al., 2011*). The film produced by the Fu catalyst is among the thickest PT films produced by cross-coupling methods (rather than electrochemical oxidative polymerization) currently reported in the literature, and this combination of catalyst and monomer may be particularly useful for preparation of polymer films for electronic devices. The film produced by the reduced PEPPSI Pd catalyst was not as thick, likely due to disproportionation of the Pt(0) catalytic center, with chain-transfer implied by the copious production of PT in solution, and confirmed to be caused by disproportionation by XPS observation of elimination of Br in the growing chain. However, the fact that the approach based on the *in situ* reduction of PEPPSI succeeded in producing a uniform film is worth reporting, as Pd(II) complexes (and PEPPSI in particular) are much easier to handle and prepare than the air-unstable Pd(0) precatalysts. To the best of our knowledge, this work is the first report in the literature of the preparation of unsubstituted PT films using PEPPSI or the popular Fu catalyst. The quantification of active catalytic chain ends by quenching the polymerization with a ferrocene probe is a useful approach, allowing the collection of kinetic data for surface-based termination processes and redox transport properties of the layer in one set of voltammetric experiments. Overall, microscopy, voltammetry, and spectroscopy of the polymerized substrates demonstrates that the spontaneously grafted aryl iodide film produced from 4-iodobenzenediazonium tetrafluoroborate is an effective and mechanically/chemically robust precatalyst for the SI-KCTP reaction.

## CONCLUSIONS

A robust aryl iodide thin film is spontaneously deposited on gold surfaces by an acetonitrile solution of 4-iodobenzenediazonium tetrafluoroborate. This is the first report of thin film formation based on an aryl halide diazonium salt which is (a) deposited from an organic solution and (b) spontaneous rather than reductively electrografted. The resulting film contains no nitrogen, in notable contrast with spontaneously deposited films from aqueous diazonium salt solutions. The aryl iodide film can be efficiently reacted with Pd(0) complexes to give surface-bound Pd(II) initiator complexes, including a complex generated *in situ* from the well-known air-stable Pd(II) cross-coupling catalyst i-Pr-PEPPSI. The surface-bound Pd(II) complex produced from the "Fu catalyst," $Pd(t-Bu_3P)_2$, initiates polymerization with 2-chloromagnesio-5-halothiophene solutions to yield densely grafted and durable polythiophene brushes on the order of 100 nm in thickness. Catalyst concentration on the surface is observed to decline over time during the polymerization, with a half-life on the order of 10 min. This decline may explain the rough morphology typically observed in SI-KCTP polymer films. Our reported initiator system is synthetically convenient both in the formation of the original aryl halide layer and its conversion into the Pd(II) surface-bound catalyst, and may find immediate use in organic electronic device construction.

## ACKNOWLEDGEMENTS

The authors thank Dr. Stavros Karakalos of the USC Columbia College of Engineering and Computing and Mr. Patrick Woodell of the Applied Research Center (Aiken, SC) for their invaluable professional services in XPS and SEM characterization respectively. The authors also thank Dr. Gerard Rowe (USC Aiken) for a generous gift of i-Pr PEPPSI catalyst complex, and Dr. Mikhail Gaevski of the Khan group (USC Columbia) for his time and assistance in profilometry measurements.

### Funding

This work was supported by a RISE grant from the Office of the Vice President for Research at the University of South Carolina Columbia. The funders had no role in study design, data collection and analysis, decision to publish, or preparation of the manuscript.

### Grant Disclosures

The following grant information was disclosed by the authors:
The University of South Carolina Columbia.

### Competing Interests

Nicholas Marshall is an Academic Editor for PeerJ.

## Author Contributions

- Nicholas Marshall conceived and designed the experiments, performed the experiments, analyzed the data, performed the computation work, prepared figures and/or tables, authored or reviewed drafts of the paper, and approved the final draft.
- Andres Rodriguez performed the experiments, authored or reviewed drafts of the paper, and approved the final draft.

## Data Availability

The raw data is available at Figshare: Marshall & Rodriguez (2020): Raw data files for SI-KCTP @ IPhN2+ Layers Peer J #43994. figshare. Dataset. https://doi.org/10.6084/m9.figshare.11420967.v2.

## Supplemental Information

Supplemental information for this article can be found online at http://dx.doi.org/10.7717/peerj-matsci.6#supplemental-information.

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
