# Peer review of "Cross-coupling polymerization at iodophenyl thin films prepared by spontaneous grafting of a diazonium salt"

_PeerJ Materials Science, doi:10.7717/peerj-matsci.6_

## Round 0.1 · original submission · Major Revisions

The reviewers have found your results of significance to the polymer community, a sentiment that I share. In particular, please pay special attention to the feedback by reviewer 2. I hope you are able to address most of their concerns.

Reviewer 1 ·

Basic reporting

The study by Marshall et al looks at the surface-initiated Kumada catalyst transfer polymerization of polythiophene after grafting of 4-iodobenzenediazonium tetrafluoroborate on gold substrates. Overall the study is interesting, well done and worthy of publication. A few minor comments should be addressed prior to publication:

Line 54. Sentence ends with “it” consider revising.

Figure 5 – more details on the CV (electrodes, solution salt concentration, etc..) should be include in the caption.

Line 163 define “PEPPSI”
Line 175 and 178: Figure 6. Figure 6 “a” should be included for clarity.
Line 178 this is the film after polymerization correct? Not before? This is not clear and should be identified.
Line 187 define “EDS”

Experimental design

good experimental design

Validity of the findings

L:114. XPS analysis: how does XPS provide thickness measurements? It provides an elemental compositional analysis. Where does the 2.5nm come from? The author should describe the assumptions used to make this analysis an provide more details (either in main text or in ESI).

For the CV (L152) can the authors comment on the stability of the films through repetitive CV measurements? Are the responses consistent after 5, 10 ,50 cycles? Is Figure 5 the first run?

Reviewer 2 ·

Basic reporting

* The Results & Discussion section would be easier to follow if a brief description of the experimental procedures used to prepare the samples were added to the beginning of each section (preparation of aryl iodide surfaces before line 112, polymerization conditions between the sentences in Line 156)

* Throughout the manuscript, there are points where added detail would make the science easier to follow, for example:
* Abstract (lines 7-8): "thick, well-defined polythiophene thin films" -- what range of thicknesses? what is meant by well-defined?
* Line 55: "the surface produced is highly variable" -- what aspect of the surface is variable?
* Line 56: "Thiol SAMs are quite reproducible" -- what aspect of the SAMs is reproducible?
* Line 131: "...coverage...consistent with a closely packed ferrocene monolayer." -- what coverage level or range is reported in the cited reference?
* Line 136: "the surface coverage...was significantly lower" -- state measure value to show how much lower.
* Line 137: "the peak-to-peak separation...indicates a slower rate" -- state the two different peak-to-peak separation values so they can be directly compared.
* Lines 146-150: "An important ancillary finding of this work..." -- This discussion is confusing. Sources for the standard procedure should be cited. What role does reaction time play? What specific method for preparing thienylmagnesium chlorides was used in this work? How much lower is the observed surface coverage if an excess of isopropylmagnesium chloride is used? Because the substrates are immersed into a solution of the Grignard reagent, presumably the concentration of thienylmagnesium chloride is much larger than the concentration of surface-bound aryl halides regardless of the preparation method used. Is the implication that the excess isopropylmagnesium chloride can also react with the surface-bound aryl halides? Were any ratios of iPrMgCl:FcCH2ThBr between 2 and 3 examined to find the lowest ratio where complete conversion could be observed (presumably this would lead to the highest yields)?
* Line 165: "highly uniform" -- How was the uniformity of the film assessed?
* Line 180: "low-polydispersity" -- How was the dispersity of the particles assessed?
* Line 187: "the brush thickness on the 2D surface is close to the particle radius." -- How were thickness and particle radius assessed? How close were the values to each other?
* Fig 6b is described as showing "a regular distribution of narrow-size distribution PT particles" -- how were the regularity of the distribution and the dispersity of the particles assessed?
* Line 205: "The film produced by the reduced PEPPSI Pd catalyst was not as thick nor the reaction as controlled..." -- What is meant by "controlled" here? Does it refer to molecular weight? dispersity? fraction of chains bound to the surface rather than in solution? A clearer description would be helpful.

* Figure 3: PEPPSI is a catalyst (Pd + ligand), not a ligand, so the definition of "L" is not accurate
* Figure 5: The caption should specify that the sample used for the CV in part (b) is on an ITO surface whereas the one in part (a) is on a gold surface.
* Lines 154-156: "Reaction...with surface-bound...catalysts has been established as a state-of-the-art technique" -- References should be cited (or re-cited) here.
* Line 174: Extraneous "(C)"?
* Figure 6: The caption for Figure S1 describes the film in Figure 6b as being cracked, but that is not mentioned in the caption to Figure 6.
* Figure 7: The caption for Part B does not appear to be correct.
* Lines 129-130/216: The ferrocene-functionalized thienyl Grignard reagent is described as being "well-known" earlier in the text, but in the experimental section the reader is referred to a prior publication (20% yield in 4 steps from dibromothiophene) from the author.

* Figure S1: The caption refers to Figure 7b (IR spectra) instead of 6b. The main conclusion from these images seems to be that there is no Au/S/C/Pd in the cracked area of the film.
* Figure S2: It is difficult to tell what conclusion is being supported by the element map--is there supposed to be a correlation between areas of S & C density? Dots corresponding to Au seem most abundant--would this be expected for a thick film of polymer? In the spectrum at the bottom, it would be useful if the region where N atoms were expected to appear were labeled so that the loss of nitrogen during grafting can be easily confirmed.
* Figure S3: The text cites this figure as evidence that a highly uniform film was prepared. It is difficult to tell from the SEM image, but it could also be interpreted as being made up of smaller fibrillar units that do not look particularly regular. Some more explanation in the text would help.
* Figure S4: This figure is confusing. The starting material (SM) should be identified as FcCH2ThBr. What was the TLC solvent system? The intensity of the baseline spot seems to increase with reaction time--is this important? The ratio of reagents should be clearly defined (equivalents of iPrMgCl to FcCH2ThBr).

Experimental design

The experiments are mostly well-designed, but the absence of any characterization data for the key molecule, 4-iodobenzenediazonium tetrafluoroborate, is very concerning. Basic characterization data should be included.

Additionally, there are several points where control experiments would be useful for supporting the conclusions:
* Line 121: "XPS of the spontaneously grafted film is consistent with a thin layer of C sp2 atoms with some adventitious carbon, including residual carbonate from oxidative cleaning at 288.5 eV." - XPS of the substrate after cleaning but before grafting would be useful to confirm the source of the sp2 C signal. (Also, Figure 4c should be be referred to here.)
* Lines 134-136: It is not clear that using trimethoxysilane grafting to an ITO surface should be directly comparable to benzenediazonium salt coupling to a gold surface. Is there any literature to support the idea that they should have comparable levels of surface coverage? There are also likely to be differences in efficacy of coupling to in iodophenyl surface (diazonium/Au system) vs a bromophenyl surface (silane/ITO system). Controls with a bromophenyldiazonium salt/Au surface and/or with an appropriate iodophenyl thiol/Au surface would be more likely to reveal more about these systems.
* Line 195: XPS (or other analysis of the film) after sonication and chloroform extraction would confirm that the film remained intact after the particle removal process.

Line 263: The "rigorous cleaning" procedure is described here to be extraction with boiling chloroform, but in the main text (Lines 194-195), the procedure is described as including both sonication in acetone and chloroform extraction.

Validity of the findings

The presentation of the findings is generally reasonable, but it leaves open a number of questions:
* Line 114: "thickness of the thin film of 2.5 nm, consistent with other reported spontaneously grafted films." A monolayer of iodophenyl groups should be much thinner than 2.5 nm (< 1 nm)--can the author comment on the accuracy of this value and what it implies about surface coverage if it is accurate? References should be cited here for other reported films.
* Lines 115-116: The measured 1:8 I/C ratio is significantly lower than the expected 1:5 ratio (implying that only 62.5% of phenyl rings still have an iodine that can participate in grafting reactions). Can the authors comment on what this would imply about potential surface coverage?
* The studies with the ferrocene probe are used to suggest the formation of a closely packed monolayer--how does this finding relate to the later conclusions about polymer surface coverage?
* Line 159: How were sample colors determined and compared? Can the color change be influenced by the gold surface? Citing absorption spectra here would be helpful and more quantitative.
* Lines 179-182: "A reasonable explanation for these...features is that they were formed from Pd(O) catalyst centers that were physisorbed onto the substrate surface" -- If this is true, then it should also happen in the ferrocene studies--what effect would excess Pd catalyst have on the surface coverage studies? Could physisorbed catalyst react with thienylmagnesium chloride and/or thienyl bromide to either give detectable solution species or excess surface-bound ferrocene?
* Lines 183-185: "[upon precipitation] the active catalyst site was inaccessible and polymerization effectively ceased." -- Is this necessarily true? This would depend upon the porosity of the particles formed upon polymer precipitation, the location of catalyst sites (particle interior vs particle surface), and solubility of monomer in polymer particle. The intended initiation sites are also bound to an insoluble material (the gold substrate), so it is difficult to argue that polymerization will occur from the gold surface and not from a polymer particle.
* Lines 187-188: "the brush thickness on the 2D surface is close to the particle radius, implying that the features may have grown as part of the same process." -- It is not clear why this would be true. By the authors' arguments, the thickness of the brush should depend upon polymer growth rate, while the particle size should depend upon whatever stage of agglomeration the precipitating solution polymer chains (which can grow from both ends and should have a different degree of polymerization than the surface bound chains) achieve by the time they adsorb to the substrate--these seem to be two very different factors.
* Lines 196-198: "...microscopy of the substrate...reveals a roughly 1 micron thick, regular film of polythiophene..." -- If the assumption is that each surface-bound aryl iodide results in one polymer chain and that there is a high density of aryl iodide sites, does a thickness of 1 micron fit with accessible molecular weights for polythiophene prepared by Pd catalysts? My instinct is that preparing polythiophene chains with a chain length of around 1 micron would mean reaching much higher molecular weights than can typically be prepared by these methods.
* Line 232: The yield for the key molecule 4-iodobenzenediazonium tetrafluoroborate is described as "0.090 g, 28%", but 0.090 g of the product (317.82 g/mol) corresponds to 0.283 mmol of product, which, starting from 2.5 mmol of 4-iodoaniline, seems to correspond to an 11.3% yield. The absence of any additional characterization data is concerning.

Additional comments

This manuscript describes some potentially interesting results, but there is significant room for improvement in the interpretation and presentation of the results. All of the areas of concern seem addressable if the authors are allowed sufficient time for revisions.

Reviewer 3 ·

Basic reporting

The paper is well written and well referenced. Figures and table are good.

Experimental design

The experimental design is presented in detailed procedures. Methods are described in detail and information will be easy to replicate.

Validity of the findings

The findings are supported by experimental evidence.

Additional comments

This manuscript reports the synthesis of polythiophene films by surface-initiated Kumada cross-coupling polymerization. The experimental methods are well described and the analysis of the polymer film by SEM is proper because it allows the determination of the film thickness. I appreciate the originality of the reported work but I do not see much potential in terms of potential applications. By using non-substituted thiophene as the monomer, there is no regioregularity problem.

The authors should determine the optical bang gap from the UV-vis spectrum of the film shown in Figure 7. From the manuscript, it is not clear if the polythiophene film thickness can be controlled. The authors should discuss if the film thickness can be controlled because this is very useful.

The authors could try to adjust the method to grow the polythiophene film on transparent ITO which will allow the characterization of the film to determine optical and electronic properties. The molecular weight of the polythiophene cannot be determined because non-substituted polythiophene is insoluble in organic solvents. I would see a lot more value to this work if 3-hexylthiophene would be used as the starting monomer to generate P3HT. This would also allow a discussion of regioregularity.

The authors recognize the formation of polymer in solution which is physisorbed on the surface of the film. The authors should try to analyze the films using TMAFM analysis to determine the rms. The morphology of the film is important, and TMAFM analysis would give some answers regarding the surface morphology of the polythiophene films.

The authors could try to measure the conductivity of the film upon doping with iodine using a four point probe.

---

## Round 0.2 · accepted · Accept

Thank you for making all those corrections. I apologize for our system being so onerous. I wish you the best and congrats to you and your coauthors.